

# Physically based summer temperature reconstruction from ice layers in ice cores

Koji Fujita[1], Sumito Matoba[2], Yoshinori Iizuka[2], Nozomu Takeuchi[3] and Teruo Aoki[4,5]

[1]Graduate School of Environmental Studies, Nagoya University, Nagoya 464-8601, Japan
[2]Institute of Low Temperature Science, Hokkaido University, Sapporo 060-0810, Japan
[3]Graduate School of Science, Chiba University, Chiba 263-8522, Japan
[4]National Institute of Polar Research, Tokyo 190-8518, Japan
[5]Meteorological Research Institute, Japan Meteorological Agency, Tsukuba 305-0052, Japan

*Correspondence to*: Koji Fujita (cozy@nagoya-u.jp)

**Abstract.** Previous studies reconstructing summer temperature from an ice core relied on statistical relationship between melt feature and instrumental temperature observed at a nearby station. This study demonstrates a novel method to reconstruct summer temperature from ice layer thickness in an ice core using an energy balance model, in which heat conduction and refreezing of meltwater in firn are taken into account. Using seasonal patterns of the ERA-Interim reanalysis dataset for an ice core site, we calculated amounts of refreezing water within firn under various settings of summer mean temperature (SMT) and annual precipitation, and prepared lookup tables containing these three variables. We then estimated SMT from the refreezing amount and annual accumulation, both available in an ice core. We applied this method to four ice cores drilled in the sites of different climates; two sites on the Greenland Ice Sheet, one in Alaska, and one in Russian Altai Mountains. Reconstructed SMTs show comparable variations to the temperatures observed at nearby stations. Relationships between SMT and ice layer thickness differ site by site, indicating that a single approximation cannot be applicable to estimate SMT. Sensitivity analyses suggest that annual temperature range, amount of annual precipitation and firn albedo setting significantly affect the relationship between SMT and ice layer thickness. This new method provides alternative and independent estimation of SMT from ice cores affected by melting.

## 1 Introduction

Ice cores drilled on alpine glaciers and polar ice sheets have provided numerous environmental information to better understand climate change. In particular, water stable isotopes have been utilized as a proxy of temperature, which is one of the most fundamental information contained in ice (e.g. Jouzel et al., 1997; Johnsen et al., 2001). However, the isotopic temperature has been a matter of discussions and investigations because the isotope information in ice was biased by the seasonality and amount of precipitation as well as sources of water vapor (e.g. Steig et al., 1994; Pfahl and Sodemann, 2014). Furthermore, because relationship between temperature and water stable isotopes varies from region to region, it is necessary to establish a relationship at the ice core site to reconstruct temperature from the water stable isotopes in the ice core. On the





other hand, ice layer (or called melt feature) has been widely used as an alternative proxy of temperatures in summer (e.g., Koerner, 1977; Herron et al., 1981; Koerner and Fisher, 1990; Fisher and Koerner, 1994). These studies have suggested warming trend based on an increase of ice layer thickness in ice cores, but a few studies quantitatively reconstructed temperature itself (e.g. Alley and Anandakrishnan, 1995; Kameda et al., 1995; Henderson et al., 2006; Okamoto et al., 2011).

In general, temperatures have been converted from ice layers by an approximation formula (linear in many cases) established between temperatures observed at nearby stations and ice layers. However, there is no guarantee that the relationship between temperature and ice layer persists over time. Since the ice layer is formed by melting and refreezing, its amount is directly resulted from the heat balance on the glacier/ice-sheet surface. In this study, we therefore aim to establish a novel approach to reconstruct summer temperature from ice layers based on a physical background. We first calculated relationship

between summer mean temperature (SMT) and ice layer using an energy mass balance model with a reanalysis climate data. SMTs were then inversely estimated from the relationship. We applied this method to four ice cores drilled at climatically different sites in the Northern Hemisphere. We also performed sensitivity analyses to understand what climatic variables affect the relationship between SMT and ice layer.

## 2 Methods and data

**2.1 Energy balance model**

We utilized an energy mass balance model (GLacIer energy Mass Balance model: GLIMB) which has been originally developed for a Tibetan glacier (Fujita and Ageta, 2000; Fujita et al., 2007). The model calculates conductive heat and change in temperatures in the glacier firn and ice, and then refreezing of percolated meltwater, which played significant role in the mass balance of cold glacier in the central Tibet (Fujita et al., 1996, 2000). GLIMB has been utilized to glaciers

distributed in different climates over the high mountain Asia (e.g., Sakai et al., 2010, 2015; Fujita et al., 2011; Fujita and Nuimura, 2011; Zhang et al., 2013). The model solves surface energy balance as:

$$\max[0; Q_m] = (1 - \alpha_s)R_s + \varepsilon R_l - \varepsilon\sigma(T_s + 273.15)^4 + H_s + H_l + Q_g \tag{1}$$

here $Q_m$ is heat for snow melting (W m$^{-2}$), $\alpha_s$ is surface albedo (dimensionless), $R_s$ is downward shortwave radiation (W m$^{-2}$)

$^{2}$), $R_l$ is downward longwave radiation (W m$^{-2}$), $\varepsilon$ is emissivity (assumed to be 1, dimensionless) for longwave radiations. $\sigma$ is the Stefan-Boltzmann constant ($5.67 \times 10^{-8}$ W m$^{-2}$ K$^{-4}$) for upward longwave radiation emitted from the surface with temperature of $T_s$ (°C). Sensible ($H_s$) and latent ($H_l$) turbulent fluxes (W m$^{-2}$) are estimated by bulk methods. The conductive heat ($Q_g$) is calculated with the surface and snow temperatures as described in the next section. All components are positive when heat fluxes are directed toward the surface. Downward longwave radiation ($R_l$) is calculated by an empirical equation





proposed by Kondo (1994), in which air temperature, relative humidity and solar radiation are taken into account. Meltwater is generated when the heat for melting is greater than zero as:

$$M_s = \frac{t_d Q_m}{l_m}$$  (2)

here $M_s$ is daily meltwater (kg m$^{-2}$ day$^{-1}$ or mm w.e. day$^{-1}$), $t_d$ is time for a day (86400 sec), and $l_m$ is latent heat of fusion

of ice (3.33 × 10$^5$ J kg$^{-1}$) daily meltwater, respectively. Since solar radiation is the main heat source for snow melting, estimation of albedo significantly affects meltwater and thus refrozen water. We utilize a scheme proposed by Kondo and Xu (1997) to estimate temporal change of daily surface albedo ($\alpha_d$, dimensionless), which is calculated as:

$$\alpha_d = (\alpha_{d-1} - \alpha_f)e^{-1/k} + \alpha_f$$  (3)

here $\alpha_{d-1}$ and $\alpha_f$ are albedo of the previous day and albedo of firn, respectively. The number of days after the latest fresh snow date is set to zero ($d = 0$) when snowfall is greater than a threshold amount ($P_{S\_min}$, mm w.e.). The surface snow albedo reduces exponentially with time. The parameter expressing reduction of albedo ($k$) depends on air temperature of ($T_a$, °C) as:

$$k = \max\left[k_{min}; k_{min} + \frac{dk}{dT}(T_a - T_t)\right]$$  (4)


here $k_{min}$, $T_t$, and $dk/dT$ are the minimum value of $k$, a threshold air temperature resulting in $k_{min}$, and a slope with negative value (< 0 °C$^{-1}$) at temperature lower than $T_t$, respectively. The albedo of fresh snow ($\alpha_0$, dimensionless) also depends on air temperature when snowfall occurs as:

$$\alpha_0 = \alpha_{max} \qquad\qquad [T_a < T_{min}]$$
$$\alpha_0 = \frac{(\alpha_f - \alpha_{max})(T_a - T_{min})}{(T_{max} - T_{min})} + \alpha_{max} \qquad\qquad [T_{min} \leq T_a \leq T_{max}] \qquad (5)$$
$$\alpha_0 = \alpha_f \qquad\qquad [T_a > T_{max}]$$


here $\alpha_{max}$ is albedo of fresh and cold snow, $T_{min}$ and $T_{max}$ are threshold air temperatures for albedo of the falling snow, respectively. Surface albedo is affected by the darker firn when the snow layer is thin. Assuming penetration of solar radiation through snow layer by Fick's second law of diffusion (Giddings and LaChapelle, 1961), the surface snow albedo ($\alpha_s$) over the underlying firn surface ($\alpha_f$) is calculated as:






$$\alpha_s = [2 - w(1-y)]/[2 + w(1-y)]$$

$$w = 2(1-\alpha_d)/(1+\alpha_d)$$

$$y = \frac{[2(1-\alpha_f) - w(1+\alpha_f)]e^{-\mu_s D_s}}{[-w(1+\alpha_f)\cosh\mu_s D_s - 2(1-\alpha_f)\sinh\mu_s D_s]}$$

(6)

here $D_s$ is thickness of snow layer above the firn surface (m), and $\mu_s$ is extinction coefficient of snow (m$^{-1}$) (Greuell and Konzelmann, 1994).

Change in snow thickness above the firn, which affects surface albedo as described in Eq. (6), is calculated as changes in

snow density at a daily time step as:

$$\frac{1}{\rho_z}\frac{d\rho_z}{dt} = \frac{\sum \rho_z dz}{\eta_z}$$

(7)

here $\rho_z$ is density of snow (kg m$^{-3}$) at depth of $z$ (m). This change is calculated with the overburden load from the depth $z$ to the surface ($\sum \rho_z dz$, kg m$^{-2}$), and viscosity of snow $\eta_z$ (kg m$^{-2}$ day), respectively. The viscosity of snow is described as:


$$\eta_z = f_w \eta_c e^{c_d \rho_z}$$

(8)

here $\eta_c$ and $c_d$ are constant parameters (kg m$^{-2}$ day and m$^3$ kg$^{-1}$, respectively), and $f_w$ is a parameter expressing water existence, which is set as 1.0 when no water exists (Fujita, 2007). Value of these parameters was determined by the Monte Carlo simulation described in the Sect. 2.3.

**2.2 Heat conduction and water refreezing**

Heat conduction in the snow/firn, which is required to estimate the surface temperature and water refreezing, is calculated with change in temperature profile within the snow/firn as:

$$\rho_z c_i \frac{\partial T_z}{\partial t} = K_s \frac{\partial^2 T_z}{\partial z^2}$$

(9)

here $\rho_z$ is density of snow (kg m$^{-3}$), which changes along depth from the surface, $z$ (m). $c_i$ is specific heat of ice (2100 J kg$^{-1}$ K$^{-1}$), $T_z$ is temperature of snow (°C) at a given depth of $z$, and $K_s$ is thermal conductivity of snow (W m$^{-1}$ K$^{-1}$), which is obtained as a function of snow density (Mellor, 1997) as:



$$K_s = 0.029(1 + 10^{-4}\rho_z{}^2) \tag{10}$$

The conductive heat is then obtained by temperature gradient at the uppermost layer of snow ($\mathrm{d}T_s/\mathrm{d}z$, K m$^{-1}$) as:

$$Q_g = K_s \frac{\mathrm{d}T_s}{\mathrm{d}z} \tag{11}$$

In this model, we calculate the temperature profile ($\mathrm{d}T_s = T_{z=0.1\,m} - T_s$) in snow at a 0.1-m interval ($\mathrm{d}z = 0.1$ m). When the surface is wet with melt or rain water, the heat transfer in snow becomes zero. Time step for calculating heat conduction is

one-hour.

Amount of refrozen water ($W_r$, kg m$^{-2}$), which forms ice layer, is calculated in three cases; 1) water percolates into cold snow layer, 2) water retained in snow is refrozen by cold snow underneath, and 3) water retained in snow is refrozen by cold events or coming winter. The case 1 can be described with the temperature of cold snow and the amount of penetrated water as:


$$W_r = \min\left[-\frac{\rho_z c_i T_z \mathrm{d}z}{l_m}; W_z\right] \tag{12}$$

here $W_z$ is percolated/retained water (kg m$^{-2}$) at a given snow layer at the depth of $z$ (m). Both cases 2 and 3 are described as the heat transfer between wet and dry (cold) layers as:

$$W_r = \min\left[-t_d K_s \frac{T_{z\prime}}{l_m \mathrm{d}z}; W_z\right] \tag{13}$$


here $T_{z\prime}$ is the temperature of a given snow layer contacting the wet snow layer at the depth of $z$, below (case 2) or above (case 3). If the wet layer is located just below the surface, $T_{z\prime}$ is replaced by $T_s$. We assume that each snow layer can retain water with a volume content $w_c$ (%), and the exceeded water percolates into the next lower layer. We also assume that the water refreezing alters the snow density up to the ice density ($\rho_i$, 900 kg m$^{-3}$), but does not prevent the water percolation. In

the calculation, we do not distinguish but sum up the amounts by these three cases as refrozen water. Density and thickness of snow change temporally based on the snow densification process, in which snow viscosity and overburden load at a given snow layer were taken into account. Details of the model scheme are described in Fujita and Ageta (2000), Fujita et al. (2007), and Fujita and Sakai (2014).





### 2.3 Model calibration

We calibrated parameters in the model with meteorological data of the Sigma-A site in northwest Greenland (Fig. 1), at which the observation has been continued since July 2012 (Aoki et al., 2014; Niwano et al., 2015, 2018). We performed the Monte Carlo simulation ($n$ =10,000) by changing the parameters related to energy and mass balance (Table S1). We determined the model parameters by comparing the simulated and observed surface level and albedo for the period from 2012 to 2015.

### 2.4 Estimation of summer mean temperature

### 2.4.1 Calculation procedure

We used daily values of meteorological variables, which are air temperature, wind speed, relative humidity, solar radiation and precipitation, from the ERA-Interim reanalysis dataset (Dee et al., 2011) as input data to create relationship among summer mean temperature, annual precipitation and refrozen water. Air temperature at elevation of ice core site was

retrieved from pressure level air temperature and geopotential height by following a manner used by Sakai et al. (2015).

To reconstruct summer mean temperature (SMT, June, July and August) from ice core, we first created look-up tables for each ice-core site, in which annual precipitation, SMT and refreezing water are summarized. To create the table, we first prepared input daily air temperature and precipitation of a given year as:

$$T_d = T_e - T_E + T_C$$
$$P_d = \frac{P_e}{P_E} P_C$$

(14)


here $T$ and $P$ denote air temperature and precipitation, respectively. Subscripts $d$ and $e$ denote daily variables of the calculation input ($d$) and those of ERA-Interim ($e$) while subscripts $E$ and $C$ denote averages (summer mean for $T$ and annual sum for $P$) of ERA-Interim ($E$) and controlled variable ($C$), respectively. Other variables, wind speed, relative humidity and solar radiation were used as those of ERA-Interim are. Initial firn temperature was assumed to be annual mean

air temperature from surface to 100 m in depth, and then surface energy balance and firn temperature profile were repeatedly calculated with the fixed temperature at the 100-m depth for 21 years to obtain a settled temperature profile for each combination of $T$ and $P$. Refrozen amount was then obtained under the controlled SMT ($T_C$) and annual precipitation ($P_C$), The controlled SMT was changed from −15 to +5 °C at an interval of 1 °C while the controlled annual precipitation ($P_C$) was changed from 20 mm to the maximum annual accumulation in ice core at an interval of 20 mm.

Because seasonality combination of air temperature and precipitation strongly affect snow melting and thus refreezing amount (Fujita, 2008; Sakai and Fujita, 2017), we created the look up table for each year from 1979 to 2013 (35 years). We determined SMT at a specific year with accumulation as annual precipitation and ice layer thickness as refrozen amount in



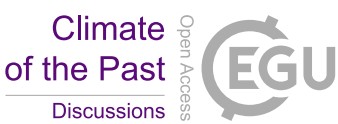

an ice core, and then estimated summer mean temperature of the specific year by averaging those of 35-year meteorological patterns.

We obtained observed refrozen amount ($W_i$, kg m$^{-2}$) from ice layer thickness ($D_i$, m) with firn density ($\rho_f$, kg m$^{-3}$) as:

$$W_i = (\rho_i - \rho_f)D_i \tag{15}$$

here we assumed the firn density before meltwater penetrating as $450 \pm 100$ kg m$^{-3}$.

### 2.4.2 Alternative estimation by calibrating ERA-Interim temperature

For cross-checking our approach addressed above, we estimated SMT by an alternative approach using the same model with the same ERA-Interim input. We first estimated a calibration parameter for precipitation ($r_p = P_I/P_E$) by comparing annual mean precipitation of ERA-Interim ($P_E$, mm) and accumulation in an ice core ($P_I$, mm w.e.), both averaged for the data available period (from 1979 to year each ice core drilled). With the calibrated precipitation, we calculated refreezing amount ($W_r$) using the same energy-mass balance model by changing input temperature ($T_d = T_e + dT_a$, °C), and then obtained root

mean square error (RMSE) and mean error (ME) against the refreezing amount estimated from the ice core ($W_i$). Considering the RMSE and ME to the calibrating temperature ($dT_a$), we determined the temperature bias for the ERA-Interim temperature to yield zero ME, and then obtained the calibrated SMT. This approach is limited by the availability of reanalysis data (since 1979 in case of ERA-Interim) so that it cannot be the main method to reconstruct SMT.

### 2.4.3 Uncertainty of the estimated summer mean temperature

We assumed that the estimated SMT has two uncertainty sources: 1) uncertainty of the estimated refreezing amount in ice core caused by the firn density assumption before the meltwater penetrating and refreezing, and 2) uncertainty caused by the look-up tables calculated with different seasonal patterns. We evaluated the uncertainty due to the density assumption ($\sigma_d$, °C) by changing the firn density as $450 \pm 100$ kg m$^{-3}$. As we demonstrate in a sensitivity analysis later, even under the same combination of annual precipitation and SMT, refreezing amount is significantly altered by the seasonal pattern of

input meteorological variables. We obtained the uncertainty due to the seasonality of input variables as standard deviation of the estimated SMTs with the 35-year patterns of ERA-Interim ($\sigma_s$, °C). The uncertainty of estimated SMT was finally obtained as a quadratic sum of the two uncertainties ($\sigma_T = \sqrt{\sigma_d{}^2 + \sigma_s{}^2}$, °C).

### 2.5 Ice cores

We applied this method to four ice cores drilled in different climatic conditions; Sigma-A site in northwest Greenland
(drilled in 2017, Matoba et al., 2018), SE-Dome in southeast Greenland (drilled in 2015, Iizuka et al., 2016, 2017, 2018; Furukawa et al., 2017), Aurora Peak in Alaska (drilled in 2008, Tsushima et al., 2015), and Mt. Belukha in Russian Altai



Mountains (drilled in 2003, Takeuchi et al., 2004; Okamoto et al., 2011; Aizen et al., 2016) (Fig. 1 and Table 1). Chronology
and net accumulation of the ice cores were determined and calibrated in respective studies. We used annual amount of
accumulation ($P_I$, mm w.e.) and ice layer thickness ($D_i$, m) in annual layer for estimating SMT. It is assumed that an ice

layer is formed with the water generated within the corresponding year. Horizontal advection and excess water infiltrated
from the succeeding layer layered above would disturb information containing ice cores even with any methods. Smoothed
density profile of each ice core was used in the model calculation. For validating the estimated summer mean temperature,
we also used temperature records observed at neighboring stations (Table 1).

**2.6 Sensitivity calculation**

Relationship between summer mean temperature and refreezing amount would be affected by a given meteorological setting,
which has been inferred from studies on mass balance sensitivity (Fujita, 2008; Sakai and Fujita, 2017). To better understand
how the relationship is governed by the given climatic condition, we performed sensitivity analyses with idealized
meteorological inputs.

**2.6.1 Idealized meteorological inputs**

We analyzed the meteorological variables of ERA-Interim used for the four ice core sites in this study (Fig. 1 and Table 1).
We first obtained a Fourier curve fitting for a given long-term averaged variable such as wind speed, air temperature and
precipitation as:

$$v_{d_y} = \bar{v} + \sum_{\kappa=1}^{2} (a_\kappa \sin \kappa d_\theta + b_\kappa \cos \kappa d_\theta) \qquad (16)$$

here $v_{d_y}$ and $\bar{v}$ are a Fourier fit variable at a given day of year ($d_y$) and annual mean, respectively. The day of year was
converted to the radian unit ($d_\theta = 2\pi(d_y/365)$). Idealized wind speed ($W_s$, m s$^{-1}$) was simply converted from the Fourier
curve fitting ($v_{d_y}$ replaced by $W_{d_y}$) by adding a Gaussian noise with the standard deviation of the ERA-Interim wind speed
($dW_s$, m s$^{-1}$) as:

$$W_s = \max \left[ W_{d_y} + dW_s; 0 \right] \qquad (17)$$


For air temperature ($v_{d_y}$ replaced by $T_{d_y}$), we added a parameter to change the annual temperature range ($R_T$, °C) as:





$$T_a = \frac{R_T}{\left|T_{d_ymax} - T_{d_ymin}\right|}\left(T_{d_y} - \overline{v_T}\right) + \overline{v_T} + \mathrm{d}T_a \tag{18}$$

here $T_{d_ymax}$ and $T_{d_ymin}$ are the maximum and minimum temperatures of the Fourier curve fitting ($T_{d_y}$, °C), respectively. $\overline{v_T}$

is the annual mean air temperature (°C), which is practically zero. A Gaussian noise with the standard deviation of the ERA-Interim air temperature (d$T_a$, °C) is also added. Normalized precipitation is defined as the daily precipitation divided by the annual sum ($P_{d+d_d}$, dimensionless), and it was described as:

$$P_{d+d_d} = \max\left[\frac{R_P}{\left|P_{d_ymax} - P_{d_ymin}\right|}\left(P_{d_y} - \overline{v_P}\right) + \overline{v_P} + \mathrm{d}P_r; 0\right]$$

$$P_r = \frac{P_{d+d_d}}{P_A} \tag{19}$$

here $P_{d_ymax}$ and $P_{d_ymin}$ are the maximum and minimum normalized precipitations of the Fourier curve fitting ($P_{d_y}$, dimensionless). $\overline{v_P}$ is the annual mean of normalized precipitation (dimensionless). Precipitation concentration can be regulated by a parameter $R_P$ (dimensionless). Precipitation seasonality, which expresses how the timing of precipitation peak differs from the standard condition, can be changed by a parameter $d_d$. A Gaussian noise with the standard deviation of the ERA-Interim normalized precipitation (d$P_r$, °C) is also added, and the obtained precipitation was set to zero if it was

negative. The idealized normalized precipitation ($P_r$, dimensionless) was finally obtained from the above mentioned normalized precipitation divided by the annual sum of $P_d$ ($P_A$, dimensionless). Air temperature and precipitation were then systematically changed to create look-up tables as described by the Eq. (14).

Relative humidity and solar radiation were parameterized with daily precipitation amount based on an analysis conducted for meteorological data at multiple sites nearby Tibetan glaciers (Matsuda et al., 2006). To obtain fitting equations, we

compared residual of relative humidity ($1 - H_r$, dimensionless) and normalized solar radiation ($R_s/R_t$, dimensionless), in which $R_t$ is the solar radiation at the top of atmosphere, with daily precipitation amount. Both variables ($1 - H_r$ and $R_s/R_t$) can be fit by exponential decay curves with precipitation (Fig. S1). Seasonal patterns of normalized variables and distributions of daily variability are shown in Fig. S2.

### 2.6.2 Variables changed

As an idealized ice core setting, we assumed an ice layer of 20 mm thickness in an annual layer of 500 mm w.e. accumulation, and then evaluated how the estimated SMT varied with given variable changes. We prepared a 30-year long daily data (Fig. S3). We tested six variables such as latitude (seasonality of solar radiation, Fig. S4), temperature range ($R_T$, Fig. S5), precipitation concentration ($R_P$, Fig. S6), and precipitation seasonality ($d_d$, Fig. S7). Also changed was firn albedo



because of its importance on heat budget at the snow surface. In addition, we tested sensitivity by changing annual
accumulation amount (Fig. S8).

## 3 Results

### 3.1 Model performance

Figure 2 shows temporal changes in surface level and albedo at the Sigma-A site, northwest Greenland. The variables
simulated are depicted (thick colored lines) as the best estimate in the Monte Carlo simulation, which was determined by the
minimum RMSE of surface level, with the following top 20 results (faint colored lines) and the observed ones (black lines).
Figure 3 shows scatter plots of the Monte Carlo simulation ($n = 10000$), for which the most correlated relationships are
selected from 16 variables (Table S1). RMSEs of surface level and albedo are highly correlated ($r = 0.571$, $p < 0.001$), and
the parameters resulting in the minimum RMSE of surface level are used for the following simulations (Fig. 3a). Most
parameters do not correlate with the RMSEs of surface level and albedo while fresh snow density ($r = -0.834$, $p < 0.001$, Fig.
3b) and firn albedo ($r = -0.738$, $p < 0.001$, Fig. 3c) significantly correlate with RMSEs of surface level and albedo,
respectively.

With the parameters yielding the best estimate of surface elevation and albedo for the Sigma-A site, we calculated RMSE
and ME of the refreezing amount by changing air temperature with the calibrated ERA-Interim precipitation at each site
(Sect. 2.4.2). There is no significant bottom on the RMSEs as shown by Fig. 4, which will be discussed in Sect.4.2. We
therefore determined a temperature bias for the ERA-Interim air temperature to yield the zero ME of refreezing amount.
Calculated refreezing amount with the biased air temperature and calibrated precipitation is compared with those estimated
from ice cores (Fig. 5). Errors shown as shadings in RMSE, ME, and refreezing amount (Figs. 4 and 5) are associated with
firn density assumption (Sect. 2.4.1 and Eq. 15).

### 3.2 Reconstructed summer mean temperatures

Figure 6 shows look-up tables, which were inversely obtained from refreezing amounts calculated with various combinations
of SMT and annual precipitation for the four ice core sites (Sect. 2.4.1). Thin dashed straight lines denote upper constraints
of the relationship between accumulation and refreezing amount, implying that given annual refreezing water could fill the
entire corresponding annual layer (with the assumed firn density of 450 kg m$^{-3}$). In another word, the annual layer fully
consists of ice. Open white circles denote refreezing amount estimated from ice layer with an assumed firn density of 450 kg
m$^{-3}$ (vertical axis), and accumulation estimated from the dated ice core as annual precipitation (horizontal axis). The tables
suggest that the relationships among SMT, annual precipitation, and refreezing amount are contrasting among the studied
sites. Accumulation amount of the Aurora ice core is so large (~1500 mm w.e., Table 1) that all the ice core data are not
shown in the look-up table (Fig. 6c). However, the look-up tables are prepared to cover the accumulation amount recorded in
the ice core.





SMTs were then finally reconstructed from the look-up tables for the four ice core sites (Fig. 7, referring left axis). Also shown are SMT anomalies recorded at nearby stations and the ERA-Interim SMTs (Table 1, referring right axis), which were biased by the alternative method (Sects. 2.4.2 and 3.1, referring left axis). It is noted that two vertical axes on both sides range the same degree of 10 °C. Two extreme cases found in ice cores: 1) annual layer fully consisting of ice (orange bars at upper part of each panel), and 2) annual layer without ice layer (blue bars at lower part of each panel) are depicted. Light

blue shading for the ERA-Interim SMT denotes an error associated with the firn density assumption, and light red shading for the reconstructed SMT denotes error associated with the assumption of firn density and multi-year seasonal patterns of input meteorological variables (Sect. 2.4.3). Relationships between the reconstructed summer mean temperature and the ice layer thickness are depicted in Fig. 8. Approximations with quadratic curves seem represent well the relationships, which are similar for Sigma-A and Aurora while it is biased to warmer side for Belukha. The SE Dome ice core contains few ice layers

and thus the approximation curve is different from those for the other sites.

### 3.3 Sensitivity to climatic variables

We conducted sensitivity analyses to understand what climatic features of input meteorological variables affect the relationship between ice layer thickness and reconstructed SMT (Fig. 9). Changes in latitude (Fig. S4), precipitation concentration (Fig. S6), and precipitation seasonality (Fig. S7) seem ineffective in the reconstructed SMT (Fig. 9a-c)

whereas annual temperature range (Fig. S5) and firn albedo largely significantly alter the SMT (Fig. 9e and 9f) even if the same setting of ice layer thickness (20 mm) and accumulation (500 mm w.e.). Also important is the annual precipitation amount (Figs. S8 and 9d).

### 4 Discussion

#### 4.1 Uncertainty in the reconstructed SMT

The reconstructed SMT includes errors derived from the firn density assumption and from the seasonal pattern of input meteorological variables. The estimated error is expressed as a quadratic sum of both errors (Sect 2.4.3), which range from 0.65 to 1.57 °C ($\sigma_T$, Table S2). Errors derived from the density assumption ($\sigma_d$) and from the seasonal pattern ($\sigma_s$) range from 0.04 to 0.15 °C, and from 0.78 to 1.57 °C, respectively (Table S2). The error derived from the density assumption would increase with ice layer thickness whereas the main error for the studied sites is caused by uncertainty of the input

meteorological variables. Our approach in this study allows quantifying errors in the temperature reconstruction, which has not been conducted in the previous studies (e.g. Henderson et al., 2006; Okamoto et al., 2011).

#### 4.2 Feasibility of the method

RMSE and ME of the calculated refreezing amount using ERA-Interim at the four ice core sites are shown in Fig. 4. There is no significant bottom on the RMSEs, which suggests that the calculated refreezing amounts are not always consistent with

the observed ice layers because zero refreezing could resulted in the flat RMSE in the cold condition. On the other hand, some periods with greater refreezing amount seem to be well represented in Sigma-A (2007-2013), Aurora (1988-1995, 2003-2006), and Belukha (1992-2002) (Fig. 5).

Look-up tables for the four ice core sites show different colour appearances (Fig. 6). Although the look-up tables for Sigma-A and SE Dome look similar, ice core data, accumulation and refreezing amount, are plotted in different domains on the

tables (Fig. 6a and 6b). The different temperature distributions imply that, even with the same ice layer thickness, the estimated SMT would differ among the sites, namely, colder in the two Greenland sites and warmer in order of Aurora and Belukha. Conversely, the same amount of meltwater (and refreezing) occurs at lower temperature in the Greenland than in Alaska and Russian Altai mountains. In the two look-up tables for Greenland, temperature isolines are almost horizontal, suggesting that the amount of annual accumulation does not affect the relationship between ice layer thickness and SMT

(Figs. 6a and 6b). On the other hand, inclined temperature isolines in the look-up tables for Aurora and Belukha suggest that, even if the same thickness of ice layer is found, reconstructed SMT will be significantly affected by annual accumulation (Figs. 6a and 6b). Namely, the reconstructed SMT will be lower in case of the lager annual accumulation.

For Sigma-A, the model successfully reproduces the warm periods since 2000 and in the 1950s whereas the temperature bottoms out in the 1960s and before the 1940s during which no ice layer frequently appears (Fig. 7a). Similar feature, no ice

layer and bottomed temperature, is also found in the SE Dome temperature (Fig. 7b) suggesting a limitation of this method. On the other hand, annual layer fully consisted of ice could have large uncertainty because excess percolated water could have been refreezing in deeper and thus older firn layer. With any proxies, no precise temperature would be reconstructed from such ice core affected by too much warming. Thick ice layer would also produce horizontal movement of percolated water in firn layer, which would disturb firn structure and preserved climatic/atmospheric information. Heavy melt event in

2012 would have disturbed the information of accumulation and refreezing in firn of the entire Greenland (e.g. Nghiem et al., 2012; Niwano et al., 2015). Although no correlation is found between the reconstructed SMTs against the temperature recorded at nearby stations and the calibrated ERA-Interim temperature, range and variability of the reconstructed and calibrated ERA-Interim SMTs are consistent each other. Uncertainty in ice core dating would directly affect the uncorrelated temperatures.

Reconstructed temperature by an empirical approximation shows little variability (SO11 in Fig. 7d) compared with those of other temperatures. This temperature was estimated using a temperature record at a nearby station (Akkem, 12 km distant from the ice-core site), one-year temperature record at the ice-core site, and ice layers in the Belukha ice core (Okamoto et al., 2011). Strong inversion layer during winter season at the Akkem station would have resulted in a warm bias in winter and then less variability on the site temperature.

Despite similar feature of the look-up tables for the two Greenland sites (Figs. 6a and 6b), direct relationships between ice layer thickness and SMT are similar between the Sigma-A and Aurora sites (Fig. 8). Significantly cold condition at the SE Dome site (Fig. 7b) and large amount of accumulation at the Aurora site (Table 1) could yield these relationships. Variation of the points around the corresponding approximate curve would be caused by variability of the annual accumulation. In



addition, if the refreezing amount can be assumed to equal to the melt amount, Fig. 8 suggests that the relationship between
SMT and melt amount can be expressed with a quadratic curve and it fluctuates with precipitation environment.

### 4.3 Parameters affecting the reconstructed SMT

Sensitivity simulation suggests that the reconstructed SMT rises as annual precipitation increases by 400 mm (Fig. 9d). More
precipitation would keep the surface albedo higher and thus warmer temperature is required to produce the same amount of
meltwater. On the other hand, the SMT falls along with the increase of annual precipitation more than 400 mm. The firn may
not be cooled enough by thick snow cover if snowfall increases and thus it could become difficult to refreeze meltwater.
As temperature range increases, opportunity for large positive temperatures during summer increases. Such warming events
would cause sufficient meltwater and thus refreezing so that the SMT itself is considered to be lower (Fig. 9e). Slight drop in
the SMT around the zero temperature range may be caused by a necessity to cool firn for refreezing.
Firn albedo as a boundary condition shows an obvious positive correlation with the reconstructed SMT (Fig. 9f). It can be
explainable by change in energy budget at the glacier surface. Lower albedo could enhance snow melting even under lower
temperature environment and vice versa. This suggests that improvement of the albedo scheme is required for more precise
estimation of SMT. In addition, to apply this method to Asian ice cores, which contain more impurities such as dust and
black carbon (Takeuchi et al., 2009; Ginot et al., 2014), we have to know how much albedo was reduced with a dusty layer
found in an ice core. Also important is a timing of dust deposition because snowfall or melting events following the
deposition would drastically alter the surface albedo (Fujita, 2007; Fujita et al., 2011).
Although it has been pointed out that precipitation concentration and seasonality strongly influenced glacier mass balance
and its response to temperature change in sensitivity analyses previously conducted (Fujita and Ageta, 2000; Fujita, 2008;
Sakai and Fujita, 2017), they do not show any significant influence on the temperature estimation (Figs. 9b and 9c). The
precipitation concentration and seasonality may not affect surface albedo at an ice core site, which is generally located or
chosen at high accumulation zone.

### 5 Conclusion

In this study, we offered a novel method to estimate summer mean temperature (SMT) from ice layers in an ice core with a
physical background. Contrasting to the traditional and empirical approach, this method allows us to estimate SMT using
information solely available in ice core without making any approximate relationship between observed temperature and ice
layer. Despite no correlation between the SMTs reconstructed by this method and observed at nearby stations, some
significant features (e.g. warm periods and recent warming) were well reproduced for the three ice core sites. At the high-
accumulation and cold site in southeast Greenland, on the other hand, the SMT was not well consistent with the
observational one because of too cold condition to produce meltwater. Applicable range of SMT by this method is likely
from –6 °C to +1 °C, and a quadratic relationship between the SMT and ice layer thickness varies under different climate
regimes. Sensitivity analyses suggest that annual temperature range and annual precipitation amount significantly affect the



relationship between the SMT and ice layer thickness, among which annual precipitation amount is available as fundamental information (accumulation rate). Firn albedo, which was assumed to be constant over time as a boundary condition, is also important to estimate summer mean temperature precisely. Improvement of the albedo scheme is required for more precise estimation.


*Data availability*. Accumulation and ice layer thickness of four ice cores used in this study are provided in the supplement.

*Author contributions*. KF designed the study, performed the simulations, analysed the data, and wrote the paper. SM (Aurora, Sigma-A, SE-Dome), YI (Sigma-A, SE-Dome) and NT (Belukha) drilled and analysed the ice core, and provided the data.
TA performed quality check of meteorological data of Sigma-A. All authors contributed to discussion of the study.

*Competing interests*. The authors declare that they have no conflict of interest.

*Acknowledgements*. T. Aoki was supported by the GCOM-C / SGLI Mission of JAXA. This study was supported by
MEXT/JSPS KAKENHI (26257201, 18H05292, 23221004, 16H01772, and 15H01733).

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



Table 1. Details of the four ice cores used in this study.

| Ice core site | Sigma-A | SE-Dome | Aurora | Belukha |
|---|---|---|---|---|
| Region | Northwest Greenland | Southeast Greenland | Central Alaska (Alaska Range) | Russian Altai |
| Latitude (°N) | 78.05 | 67.18 | 63.52 | 49.81 |
| Longitude (°) | 292.37 | 323.63 | 213.46 | 86.56 |
| Altitude (m a.s.l.) | 1500 | 3190 | 2830 | 4100 |
| Year the ice core drilled | 2016 | 2014 | 2008 | 2003 |
| Coverage year* | 1902 | 1956 | 1960 | 1914 |
| Mean annual accumulation rate of ice cores ($P_I$, mm w.e.) | 314 | 1058 | 1495 | 399 |
| Mean annual precipitation of ERA-Interim ($P_E$, mm) | 197 | 1011 | 717 | 810 |
| Precipitation ratio ($r_P$) | 1.60 | 1.05 | 2.09 | 0.49 |
| Bias for ERA-Interim temperature ($dT_a$, °C) | −1.52 ± 0.28 | +0.45 ± 0.29 | −0.37 ± 0.30 | +2.35 ± 0.27 |
| Nearby stations (distance from the ice core site, km) | Qaanaaq (93), Thule (177) | Tasiilaq (187) | Big Delta (77), Gulkana (76), Fairbanks (118) | Akkem (12) |
| References for ice core | Matoba et al. (2018) | Iizuka et al. (2016); Furukawa et al. (2017) | Tsushima et al. (2015) | Takeuchi et al. (2004); Okamoto et al. (2011) |

* Year to which ice layers are analyzed.



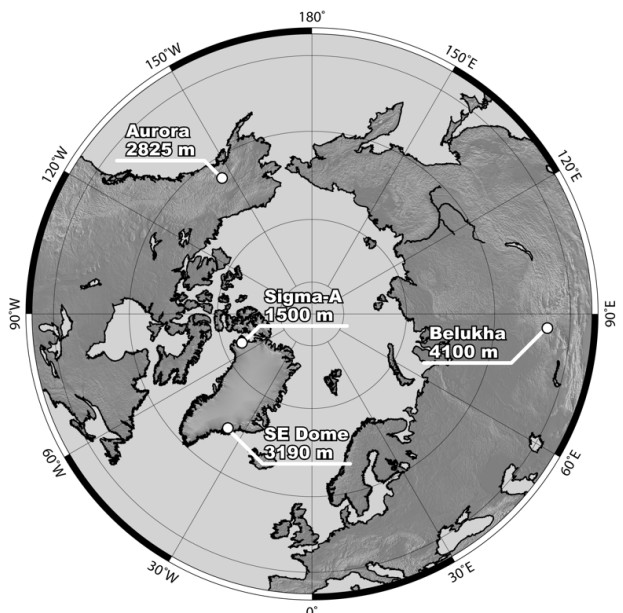

**Figure 1. Locations of the ice-core sites. Also see Table 1.**




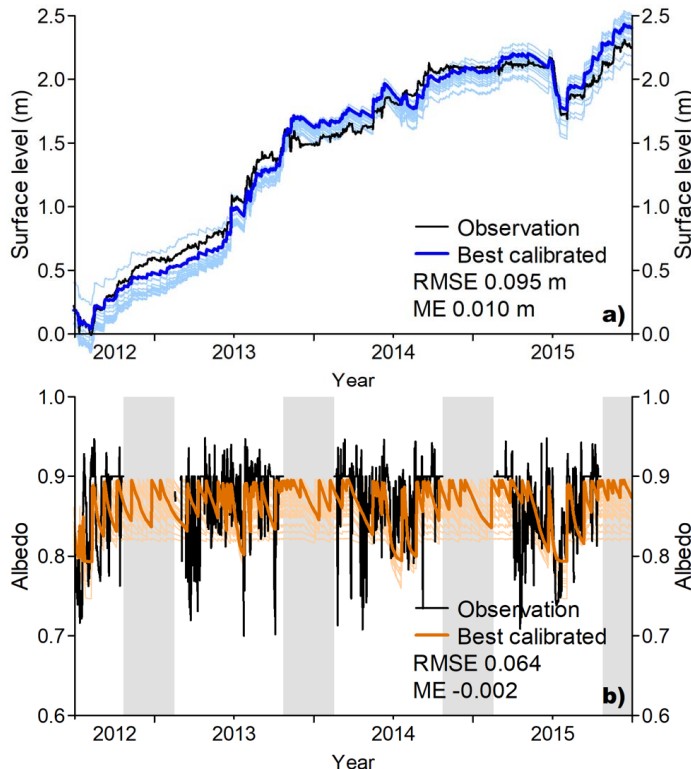

**Figure 2.** Temporal changes in (a) surface level and (b) albedo at the Sigma-A site, northwestern Greenland, from July 2012 to December 2015. The thick solid and thin coloured lines denote the best calibrated result and top 20 estimates from the Monte Carlo simulation, respectively. RMSE and ME denote the root mean square error and mean error, respectively. The grey shaded regions in (b) denote the winter (poler night) period when there were no albedo observations.



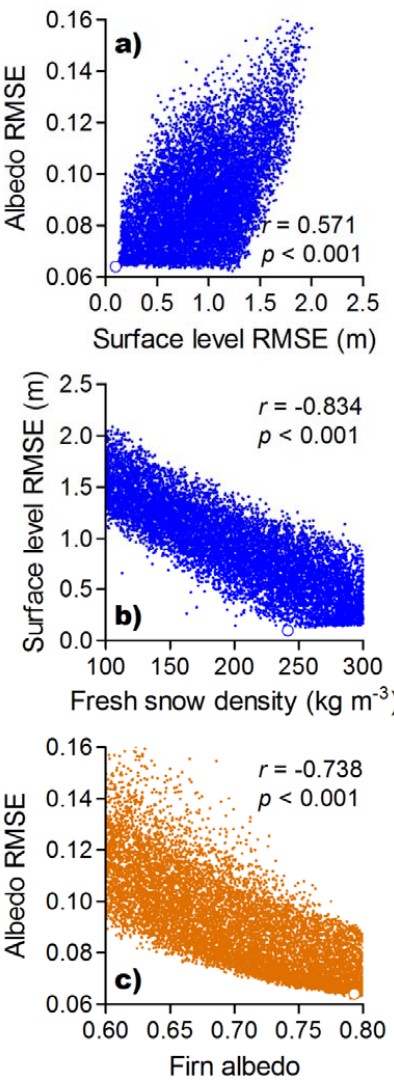

**Figure 3. Scatter plots of the Monte Carlo simulation (n = 10,000), showing the highest correlations between (a) surface level and albedo RMSEs, (b) fresh snow density and surface level RMSEs, and (c) firn albedo and albedo RMSEs. Open circles denote the parameters that yield the best estimate of the surface level RMSE.**






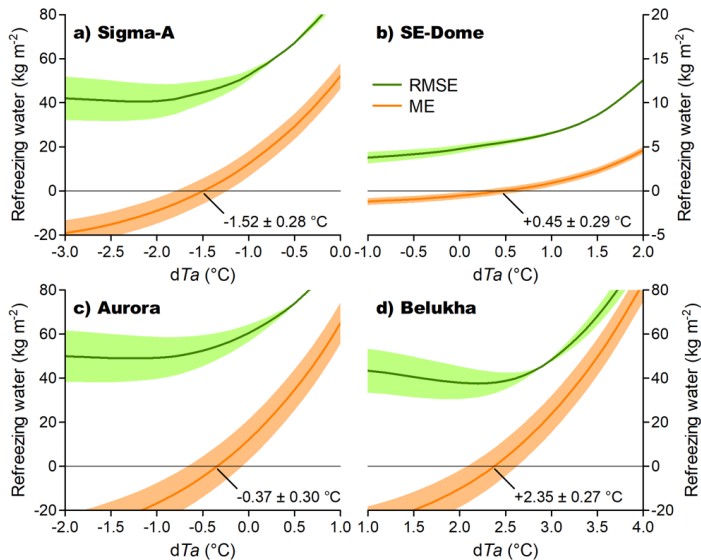

**Figure 4. RMSE and mean error (ME) of the calculated refreezing amount using the ERA-Interim dataset at the four ice-core sites. The temperature where ME equals zero was adopted as the calibration temperature to calculate the best estimate of the refreezing**

**water. The shaded regions denote the errors associated with the firn density assumption (Sect. 2.4.1).**



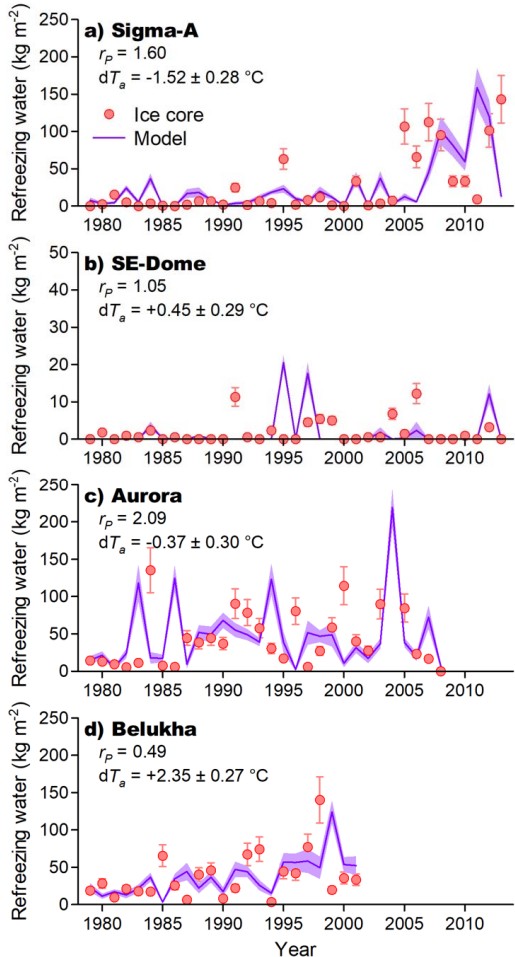

**Figure 5.** Observed (red circles, ice core) and simulated (purple lines, model) refreezing water for the four ice cores: (a) Sigma-A, northwestern Greenland, (b) SE-Dome, southeastern Greenland, (c) Aurora Peak Alaska, and (d) Mt. Belukha, Russian Altai Mountains. The error bars for the ice core and shaded regions for simulated refreezing water were derived from the firn density assumption.





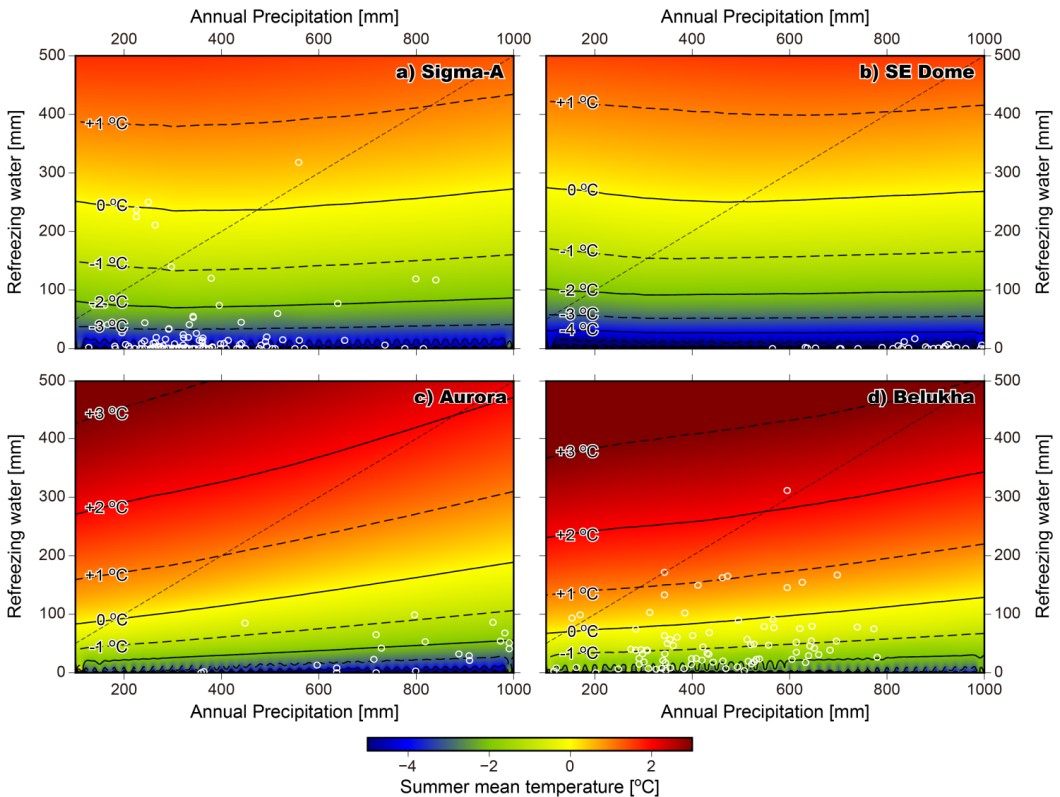

**Figure 6. Lookup tables for the summer mean temperature (SMT) against annual precipitation (horizontal axes) and refreezing**

**water (vertical axes) for the four ice-core sites: (a) Sigma-A, northwestern Greenland, (b) SE-Dome, southeastern Greenland, (c)**

**Aurora Peak, Alaska, and (d) Mt. Belukha, Russian Altai Mountains. The open white circles denote the ice-core data for each**

**respective site. The thin linear dashed lines denote the 100% melt feature percentage, which implies that one annual layer consists**

**entirely of refrozen ice for a firn density assumption of 450 kg m$^{-3}$.**

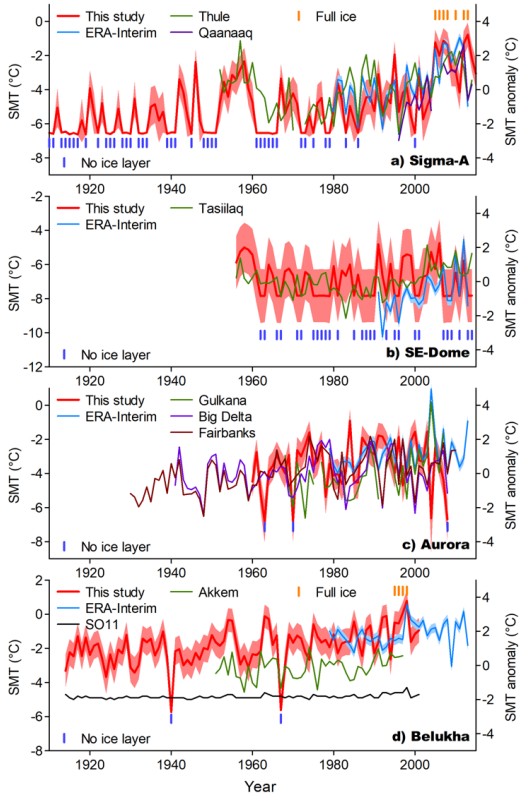


**Figure 7. Reconstructed SMTs from the four ice cores: (a) Sigma-A, northwestern Greenland, (b) SE-Dome, southeastern Greenland, (c) Aurora Peak, Alaska, and (d) Mt. Belukha, Russian Altai Mountains. The reconstructed and ERA-Interim calibrated temperatures correspond to the left y-axis, whereas the SMT anomalies of the nearby stations correspond to the right y-axis. The light-blue shaded regions for the ERA-Interim SMTs denote the error associated with the firn density assumption, and**

**the light-red shaded regions for the reconstructed SMTs denote the errors associated with the firn density assumption and multi-year seasonal patterns of the input meteorological variables. The annual layers without an ice layer (blue bars) and with fully refrozen ice (orange bars) are also shown. The SO11 model results in (d) denote reconstructed SMTs from the study of Okamoto et al. (2011).**





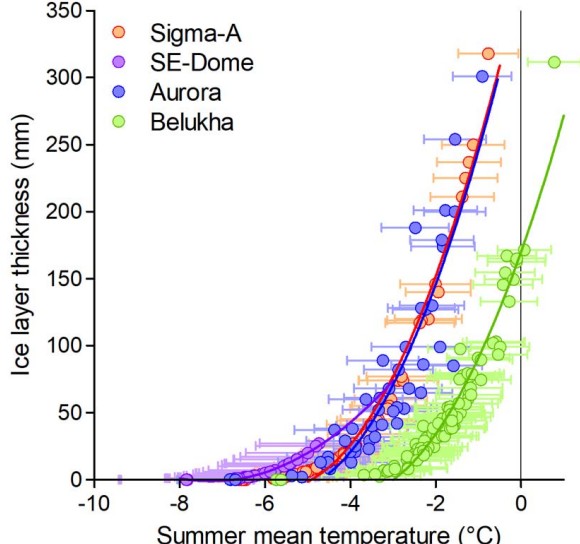


**Figure 8. Relationships between SMT and ice-layer thickness of the four ice cores. The horizontal bars denote the errors associated with the firn density assumption and meteorological patterns.**





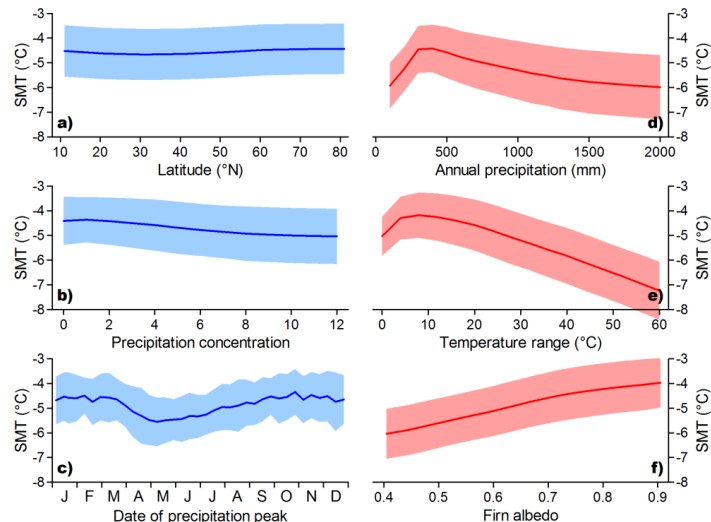

**Figure 9. Sensitivity of the estimated SMTs to the following variables: a) latitude, b) precipitation concentration, c) date of precipitation peak (precipitation seasonality), d) annual precipitation, e) temperature range, and f) firn albedo. We assumed a 20-m-thick ice layer and 500 mm w.e. annual accumulation (except for the sensitivity to (d), annual precipitation). The light-coloured shaded regions denote the estimation errors associated with the firn density assumption and year of calculation (see Sect. 2.4.3).**