# Peer review of "Physically based summer temperature reconstruction from ice layers in ice cores"

_Climate of the Past, 2019_

## Referee Comment (RC1) · Anonymous Referee #1 · 28 Sep 2019

The manuscript entitled "Physically based summer temperature reconstruction from ice layers in ice cores" by Fujita et al. presents further understanding on the method to reconstruct summer temperature from ice layer thickness using an energy balance model. Given the complex interpretations of ice core stable isotopic record as a temperature indicator, a Physically based temperature paramater holds its merits for publication. However, I concerned much about the feasibility of the method presented here. As indicated by the authors, applicable range of this method is likely from –6 °C to +1 °C, while uncertainty in the reconstructed SMT is comparatively high. Errors derived from the density assumption and from the seasonal pattern range from 0.04 to 0.15 °C, and from 0.78 to 1.57 °C, respectively (Table S2). The situation might worsen when considering uncertainty with the density assumption and the seasonal pattern.

[Figure]

Especially the firn densification model requires prior parameters on temperature and accumulation.

Ideal assumptions bear upon the energy balance model. How much uncertainty can these assumptions can bring about? Some assumptions require further confirmation, for instance, the authors assume that each snow layer can retain water with a volume content, and the exceeded water percolates into the next lower layer. The volume content might be partly dependent on the snow temperature distribution and ice layers.

minor comments: (1) Possibly better to make use of cumulative temperature in summer than SMT? (2) What if the method were performed on the ice cores that are recovered at different elevations of the accumulation zone of the same glacier? (3) Line 234: why Tibetan glaciers? (4) Line 317: Is there independent evidence to support the correlation between SMT and accumulation? (5) Line 351: Is this reasonable given the more complicate albedo scheme?

―――――――――――――――――

---

## Author Comment (AC1) · 10 Oct 2019

Dear Reviewer #1

Thank you for your comments on our manuscript submitted to CP.

[reviewer comment] The manuscript entitled "Physically based summer temperature reconstruction from ice layers in ice cores" by Fujita et al. presents further understanding on the method to reconstruct summer temperature from ice layer thickness using an energy balance model. Given the complex interpretations of ice core stable isotopic record as a temperature indicator, a Physically based temperature paramater holds its merits for publication.

[author reply] Thanks for the positive evaluation.

[Figure]

[reviewer comment] However, I concerned much about the feasibility of the method presented here. As indicated by the authors, applicable range of this method is likely from –6 âŮęC to +1 âŮęC, while uncertainty in the reconstructed SMT is comparatively high. Errors derived from the density assumption and from the seasonal pattern range from 0.04 to 0.15 âŮęC, and from 0.78 to 1.57 âŮęC, respectively (Table S2). The situation might worsen when considering uncertainty with the density assumption and the seasonal pattern. Especially the firn densification model requires prior parameters on temperature and accumulation.

[author reply] We think that this method has good feasibility for SMT reconstruction; this is because, the reconstructed temperatures showed large inter-annual variability exceeding error range except for SE-Dome site (Fig. 7). In addition, as we wrote in the introduction, a few studies have reconstructed temperature from melt features in ice cores while the other studies described just warmer/cooler. Further, those previous studies have rarely presented error range of their estimates. So, we believe that it is valuable to provide the error ranges itself even if they are large. We also believe that it is valuable to provide alternative information with this "independent method" even if its applicable temperature range is narrow.

Because meltwater refreezing occurs in the first annual layer, we think that the uncertainty of density assumption in the current manuscript is reasonable (350 to 550 kg mˆ–3, L168). Although meltwater can penetrate into deeper firn layer in the model, we do not assume such "internal accumulation" in this study (see L321). Even if the density range was twice greater than the current range (250 to 750 kg mˆ–3), the error range would be still less than that from seasonal pattern (0.08 to 0.30 degC, <20% of that due to seasonal pattern).

The greater error range due to seasonal pattern suggests that the linear relationship between melt feature and neighboring instrumental temperature, which has been used in the previous studies, would not work. We will add this assertion around L340.

Although the reviewer addressed "might worsen", uncertainty due to the seasonal pattern is estimated from the 35-year patterns so that uncertainty of the variability is already considered.

We do not catch what the last sentence means. Density profile by the densification model is not used for conversion from ice layer thickness to refreezing amount but used for temperature conduction. So, the densification scheme does not affect the density assumption for the conversion.

[reviewer comment] Ideal assumptions bear upon the energy balance model. How much uncertainty can these assumptions can bring about? Some assumptions require further confirmation, for instance, the authors assume that each snow layer can retain water with a volume content, and the exceeded water percolates into the next lower layer. The volume content might be partly dependent on the snow temperature distribution and ice layers.

[author reply] It is unfortunate that the reviewer describes just "some assumptions" without providing specific parameters. Anyway, we perform additional sensitivity test for parameters shown in Table S1. For reducing the calculation procedure, we do this test as a part of the sensitivity test with idealized meteorological input. We find that the largest SMT range is yielded by firn albedo (1.28 degC), followed by fresh snow albedo (0.29 degC), threshold air temperature for rain probability (0.28 degC), and then the minimum value of k (0.22 degC). This reasonably agrees with the importance of albedo setting concluded with the original sensitivity test (Sect. 4.3). Most of other parameters yield less than 0.01 degC. Water content does not affect SMT range (0.00 degC) even if the range was expanded to 3-10%. We will add the results to Table S1 and add the descriptions in Sect. 2.6.2 and 3.3.

[reviewer comment] (1) Possibly better to make use of cumulative temperature in summer than SMT?

[author reply] Does this mean positive degree day (PDD)? If so, we have confirmed

that the relationships between PDD and ice layer thickness (see attached figure) were almost linear. However, for the purpose of the study, we do not think that it is meaningful to provide PDDs as a temperature index instead of the summer mean temperature. We will add the description on PDD around L340 but will not provide figure.

[reviewer comment] (2) What if the method were performed on the ice cores that are recovered at different elevations of the accumulation zone of the same glacier?

[author reply] We believe that we can retrieve temperature lapse rate between the sites. We would not add any description about this in the revised manuscript.

[reviewer comment] (3) Line 234: why Tibetan glaciers?

[author reply] Because we had data and performed the study. But this is just a trigger for the analysis. We conducted the analysis with ERA-Interim data in this study (not Tibet), and found the similar relationships among these parameters.

[reviewer comment] (4) Line 317: Is there independent evidence to support the correlation between SMT and accumulation?

[author reply] Yes, that is why model-based studies were cited at L200 and L357. If this means "observational evidence", we do not have it.

[reviewer comment] (5) Line 351: Is this reasonable given the more complicate albedo scheme?

[author reply] If such "complicated albedo scheme" can reproduce the snow surface condition more realistically, it could be better for improvement. But as we addressed in L352, effects of dust and black carbon should be more significant.
* * *
[Figure]

**Fig. 1.** ice layer vs. pdd

---

## Referee Comment (RC2) · Elizabeth Thomas (Referee) · 29 Oct 2019

Ice cores have long been used to reconstruct past surface temperatures, however in areas of surface melting this is not always possible. This paper presents a new method for reconstructing summer temperatures from melt layers in ice cores. The new method, based on an energy balance model, provides a valuable alternative to traditional surface temperature proxies, however the potential limitation is that melt must be present.

General comments: I found the term "ice layer" confusing. What you are referring to is "melt layers" or even "ice lenses" that occur in the firn.

Line 127. I am unsure of the assumption "water refreezing alters the snow density up to the ice density (pi, 900 kg m–3), but does not prevent the water percolation". Can

the water percolate through the ice layers? One of my concerns with the method is that melt layers act as a barrier for further melt percolation. Thus what might appear to be a large melt layer could be comprised of several smaller melt events. In this case you would over estimate your summer temperature reconstruction. This is also a drawback of using the annual ice layer thickness (line 194). Is the assumption therefore that the melt occurs in a single event each summer?

I think you need more information about the ice core data used. Either in the text description or in table 1. How were the ice cores dated? What is the approximate dating uncertainty? Are your years from summer to summer or winter to winter? How was the ice layer thickness determined? Visual? Line scanner? How accurate are these measurements? Can you determine small melt layer or just large melt events? Is it possible to identify multiple smaller melt events? Can you identify melt layers at depth or is it only possible in the firn?

I am not sure if there is a better term for "look-up tables" but I found the term strange. Would calibration tables be better?

How well does ERA-interim capture conditions at the ice core sites? Have there been any studies to demonstrate this? My concern is that the approach is heavily dependent on the reanalysis data, but for many ice core sites (especially those subject to melt) the spatial resolution of ERA-interim may not be suitable. Is there a way you can demonstrate that ERA-interim is suitable?

I think the issue of impurities in the ice could be a limitation to this method. The authors include a caveat in the discussion that the albedo scheme needs improving. I think this is especially important for coastal or continental sites, which may be subject to local dust sources. The surface mass-balance model by Goelles and Boggild includes a dynamic ice albedo component. In addition to dust and black carbon, this model includes clouds and the angle of the sun. GOELLES, T., & BØGGILD, C. (2017). Albedo reduction of ice caused by dust and black carbon accumulation: A model applied to the K-transect, West Greenland. Journal of Glaciology, 63(242), 1063-1076. doi:10.1017/jog.2017.74

I think a new method of reconstructing temperature that is not reliant on stable water isotopes is important. However, stable water isotopes are a well-established method. I wonder if it would strengthen your case to include the stable water isotopes for these ice cores in your figures (Fig. 7), or a supplementary figure, to demonstrate the imperfect nature of the stable water isotopes – temperature relationship. I found the correlation between SMT and ERA-interim convincing but clearly it is not an exact match. However, if you presented the stable water isotopes you would also expect differences.

Is the SMT reconstruction from stable water isotopes better or worse than your method? Is it even possible to get a summer mean temperature from isotopes? I think you should include some additional background in the introduction about the drawbacks of other temperature reconstructions and how the information can be lost in the presence of surface melt. Future climate warming means we need additional methods of extracting climate information from ice cores that may be subject to melt.

---

## Referee Comment (RC3) · Anonymous Referee #3 · 11 Nov 2019

Fujita et al., Physically based summer temperature reconstruction from ice layers in ice cores

General Comments: This manuscript presents a new way to reconstruct past temperature from ice cores. Encouragingly, this method relies on the melt features that can frequently confound the estimation of temperature using more traditional methods of variations in oxygen (and potentially hydrogen) isotopes. The manuscript tests the method on a range of ice cores which are widely differentiated both geographically and in elevation. The data encapsulated in Figure 8 indeed looks very encouraging.

Line 31-32 – are there any more recent references to melt features being used to characterise temperature? Apart from a couple in the 2000's, these references are 20 or more years old. Also – using the term 'ice layer' or 'ice layer thickness' is confusing.

[Figure]

Are you talking about melt layers in the ice? Layer thickness in ice cores usually refers to annual layer thickness.

Where are your methods where you outline the analysis of the ice cores you used, and thus how they were dated. The annual layer dating of these cores is critical to this project since you are trying to reconstruct summer mean temperature.

Was there a reason to not use ERA5 rather than ERA interim? ERA 5 would provide a smaller grid size, and therefore parameters like 2m temp may be more realistic. ERA 5 is likely far more relevant to a study looking at calibrating grid data with specific ice core sites.

How were the 'ice layers' or melt layers differentiated from bubble free layers that may have formed via other means? E.g. bubble free layers in ice cores have been observed at various sites and have been differentiated from melt layers via their appearance (Feyveresi et al., 2018, The Cryosphere 12:325-341) and even via analysis of their noble gas chemistry (Orsi et al., 2015, J. Glaciology doi:10.3189/2015JoG14J237). These layers can result from surface crusts that have been retained and buried, and the surface crusts may have formed via wind scouring, or other atmospheric processes like inversions. These are not melt processes. The study needs to provide some detail of how melt layers were discerned from other layers in the cores - perhaps including some photos?

Specific Comments: I suggest some proof-reading to improve the English. There are numerous instances of missing words, e.g. first sentence of the abstract "...relied on the statistical analysis...", line 29 'because the relationship...'.

What does 'firn albedo setting' mean in the abstract? Perhaps explain briefly here.

It would be useful to provide some more detail about the nearby stations used in Table one – e.g. elevation, length of observations, not only distance but also direction from the ice core site.

Can you split the data in figure 8 into four separate graphs on the one figure? It would be preferable to see the four sites more clearly. Another option would simply be to make the x axis far longer (although still covering the same temp interval) so that it is easier to differentiate the four sites. It is hard to see the orange and pink dots.

---

## Author Comment (AC2) · 28 Nov 2019

Dear Elizabeth Thomas as Reviewer #2

Thank you for your comments on our manuscript submitted to CP.

[reviewer comment] Ice cores have long been used to reconstruct past surface temperatures, however in areas of surface melting this is not always possible. This paper presents a new method for reconstructing summer temperatures from melt layers in ice cores. The new method, based on an energy balance model, provides a valuable alternative to traditional surface temperature proxies, however the potential limitation is that melt must be present.

[author reply] Thanks for the positive evaluation.

[Figure]

[reviewer comment] General comments: I found the term "ice layer" confusing. What you are referring to is "melt layers" or even "ice lenses" that occur in the firn.

[author reply] We changed the term to "melt layers" including the title.

[reviewer comment] Line 127. I am unsure of the assumption "water refreezing alters the snow density up to the ice density (pi, 900 kg m–3), but does not prevent the water percolation". Can the water percolate through the ice layers? One of my concerns with the method is that melt layers act as a barrier for further melt percolation. Thus what might appear to be a large melt layer could be comprised of several smaller melt events. In this case you would over estimate your summer temperature reconstruction. This is also a drawback of using the annual ice layer thickness (line 194). Is the assumption therefore that the melt occurs in a single event each summer?

[author reply] Modeling percolation of large amount of meltwater is a challenging issue and we have no idea how to improve the present scheme of this study. If we have a scheme expressing that a melt layer prevents meltwater penetration, however, the reconstructed SMT would be "under estimated" because thicker layer would be formed than that by the present scheme under the same temperature condition. If multi melt layers are identified in an annual layer, we assume a single melt layer thickness by summing up thickness of those layers. This issue would cause large SMT errors especially for the period during which thick melt layers are found (orange bars in Fig. 7). We will add these descriptions in the discussion section.

[reviewer comment] I think you need more information about the ice core data used. Either in the text description or in table 1. How were the ice cores dated? What is the approximate dating uncertainty? Are your years from summer to summer or winter to winter? How was the ice layer thickness determined? Visual? Line scanner? How accurate are these measurements? Can you determine small melt layer or just large melt events? Is it possible to identify multiple smaller melt events? Can you identify melt layers at depth or is it only possible in the firn?

[author reply] We will add dating method, age markers, dating error, and method for melt layer measurement in Table 1. Annual layer of three ice-cores are defined between winters while the SE-Dome ice-core is dated in monthly scale. The minimum melt layer thickness and its accuracy are $1 \pm 1$ mm in all ice cores. If multi melt layers are identified in an annual layer, we assume a single melt layer thickness by summing up thickness of those layers. We will add these descriptions in Sect. 2.5. We only dealt with melt layers in the firn in this study, and the identification of melt layer at depth of ice is out of focus of this study. We will add a brief description about this issue in the discussion section.

[reviewer comment] I am not sure if there is a better term for "look-up tables" but I found the term strange. Would calibration tables be better?

[author reply] "Calibration table" sounds strange for us because we do not calibrate any results through the tables. Once we create a table, we can retrieve a value of the target variable (SMT: summer mean temperature in this study) from a combination of explanatory variables (annual accumulation and refreezing amount in this study). We think that this should be called "lookup tables".

[reviewer comment] How well does ERA-interim capture conditions at the ice core sites? Have there been any studies to demonstrate this? My concern is that the approach is heavily dependent on the reanalysis data, but for many ice core sites (especially those subject to melt) the spatial resolution of ERA-interim may not be suitable. Is there a way you can demonstrate that ERA-interim is suitable?

[author reply] We do not think that there are studies confirming the validity of reanalysis datasets (not only ERA-Interim but also ERA5, NCEPs, MERRA and others) at an ice core site where observational data is generally unavailable. Validity of ERA-Interim air temperature has been tested with several observational data in the high mountain Asia (Sakai et al., 2015) though they are not located at high-elevation ice core site but around glacier termini. On the other hand, representativeness of those datasets does

not matter in this study because air temperature and precipitation are systematically modified to obtain "lookup table". Our sensitivity tests show that the annual temperature range only affects the estimated SMT. This suggests that the reanalysis data would be suitable for demonstrating this study if the annual temperature range was reliable at the ice core sites even though the representativeness of temperature and precipitation amount were not precise. We will add some descriptions about this in the revised manuscript. Other effective parameters are precipitation and firn albedo. But precipitation is a given parameter from ice core and issue of firn albedo (and albedo scheme improvement) is already addressed in the original manuscript.

[reviewer comment] I think the issue of impurities in the ice could be a limitation to this method. The authors include a caveat in the discussion that the albedo scheme needs improving. I think this is especially important for coastal or continental sites, which may be subject to local dust sources. The surface mass-balance model by Goelles and Boggild includes a dynamic ice albedo component. In addition to dust and black carbon, this model includes clouds and the angle of the sun. GOELLES, T., & BØGGILD, C. (2017). Albedo reduction of ice caused by dust and black carbon accumulation: A model applied to the K-transect, West Greenland. Journal of Glaciology, 63(242), 1063-1076. doi:10.1017/jog.2017.74

[author reply] The issue of impurities has been already addressed at L352-355 of the original manuscript. The study provided here (Goelles and Bøggild, 2017) mainly deals with albedo and melting processes in the ablation zone while our study focuses on that in the accumulation zone. In addition, as we have addressed in the original manuscript, our studies have revealed that "deposition timing of impurities" is much more important than albedo schemes. We admit that our albedo scheme is not sophisticated but we do not think that the study by Goelles and Bøggild (2017) is appropriate to be cited for discussing the issue and impact of impurities.

[reviewer comment] I think a new method of reconstructing temperature that is not reliant on stable water isotopes is important. However, stable water isotopes are a

well-established method. I wonder if it would strengthen your case to include the stable water isotopes for these ice cores in your figures (Fig. 7), or a supplementary figure, to demonstrate the imperfect nature of the stable water isotopes – temperature relationship. I found the correlation between SMT and ERA-interim convincing but clearly it is not an exact match. However, if you presented the stable water isotopes you would also expect differences. Is the SMT reconstruction from stable water isotopes better or worse than your method? Is it even possible to get a summer mean temperature from isotopes? I think you should include some additional background in the introduction about the drawbacks of other temperature reconstructions and how the information can be lost in the presence of surface melt. Future climate warming means we need additional methods of extracting climate information from ice cores that may be subject to melt.

[author reply] We compare our SMT and deuterium stable water isotope (SWI). Figure 1, which will be shown as a supplementary figure in the revised manuscript, shows that inter-annual variabilities of deuterium are different site by site; large at the Sigma-A and Belukha sites while small at the SE-Dome and Aurora sites. In addition, fluctuation and trend of SWI are different from those of SMT at the same site. This is probably because the annual SWI signal is affected by winter accumulation and seasonal variability of precipitation. In addition, the SWI approach requires observational temperature data to convert SWI to temperature, which is the same issue in the empirical approach using melt layers. We will add descriptions about this comparison in the discussion section of the revised manuscript. We add two colleagues by contributing these isotope data.

———————————————————

[Figure]

**Fig. 1.** Reconstructed (red lines) and ERA-Interim (light blue) summer mean temperature (SMT, left axes), and deuterium water stable isotope of ice core (light green, right axes)

---

## Author Comment (AC3) · 28 Nov 2019

Dear Reviewer #3

Thank you for your comments on our manuscript submitted to CP.

[reviewer comment] General Comments: This manuscript presents a new way to reconstruct past temperature from ice cores. Encouragingly, this method relies on the melt features that can frequently confound the estimation of temperature using more traditional methods of variations in oxygen (and potentially hydrogen) isotopes. The manuscript tests the method on a range of ice cores which are widely differentiated both geographically and in elevation. The data encapsulated in Figure 8 indeed looks very encouraging.

[Figure]

[author reply] Thanks for the positive evaluation.

[reviewer comment] Line 31-32 – are there any more recent references to melt features being used to characterise temperature? Apart from a couple in the 2000's, these references are 20 or more years old.

[author reply] We have checked papers including keywords ["ice core" and "temperature" and "reconstruction"] in Google Scholar for the period 2011-2019 (results ~17,200). Among the first 150 papers sorted by relevance (after these, conference abstracts increase), we exclude studies using tree ring or sediment core. And we found that 85 papers use isotopic approach (including papers using existing data), seven papers use borehole temperature, one paper deals with gas analysis, four papers use other proxies (pollen, bubble and ELA), and TWO papers adopt ice-layer approach. Furthermore, one of the two papers is our study (Okamoto et al., 2011, JGR) which has been already cited in this study. So, we could find only one study in Antarctic Peninsula by Abram et al. (2013, Nature Geosci.). We will add this study to our revised manuscript.

[reviewer comment] Are you talking about melt layers in the ice? Layer thickness in ice cores usually refers to annual layer thickness.

[author reply] We changed the term to "melt layers" including the title.

[reviewer comment] Where are your methods where you outline the analysis of the ice cores you used, and thus how they were dated. The annual layer dating of these cores is critical to this project since you are trying to reconstruct summer mean temperature.

[author reply] Although it is difficult to understand what the first sentence intends to mean, we imagine that you request more details of the ice cores and their dating method. Dating methods are different ice-core by ice-core, which are described in publications listed in Table 1. We will add brief descriptions and add information to Table 1 (dating method and error).

[reviewer comment] Was there a reason to not use ERA5 rather than ERA interim? ERA 5 would provide a smaller grid size, and therefore parameters like 2m temp may be more realistic. ERA 5 is likely far more relevant to a study looking at calibrating grid data with specific ice core sites.

[author reply] This is because we have easy access to the ERA-Interim data (already downloaded and modified to daily data). In addition, there is no way to confirm which dataset (including NCEPs, MERRA and others) is "realistic" at ice core sites where observational data is generally unavailable. Validity of ERA-Interim air temperature has been tested with several observational data in the high mountain Asia (Sakai et al., 2015) though they are not located at high-elevation ice core site but around glacier termini. However, representativeness of those dataset does not matter in this study because air temperature and precipitation are systematically modified to obtain "look-up table". Our sensitivity tests show that the annual temperature range only affects the estimated SMT. This suggests that the reanalysis data would be suitable for demonstrating this study if the annual temperature range was reliable at the ice core sites even though the representativeness of temperature and precipitation amount were not precise. We will add some descriptions about this in the revised manuscript. Other effective parameters are precipitation and firn albedo. But precipitation is a given parameter from ice core and issue of firn albedo (and albedo scheme improvement) is already addressed in the original manuscript.

[reviewer comment] How were the 'ice layers' or melt layers differentiated from bubble free layers that may have formed via other means? E.g. bubble free layers in ice cores have been observed at various sites and have been differentiated from melt layers via their appearance (Feyveresi et al., 2018, The Cryosphere 12:325-341) and even via analysis of their noble gas chemistry (Orsi et al., 2015, J. Glaciology doi:10.3189/2015JoG14J237). These layers can result from surface crusts that have been retained and buried, and the surface crusts may have formed via wind scouring, or other atmospheric processes like inversions. These are not melt processes.

[Figure]

The study needs to provide some detail of how melt layers were discerned from other layers in the cores - perhaps including some photos?

[author reply] Because this study does not aim to develop a method identifying melt layer, we have no idea about this comment. We will add some descriptions addressing that those "melt-layer-like" layers would affect the estimated SMT but those layers would be thinner than melt layer.

[reviewer comment] Specific Comments: I suggest some proof-reading to improve the English. There are numerous instances of missing words, e.g. first sentence of the abstract "...relied on the statistical analysis...", line 29 'because the relationship...'.

[author reply] Though the original manuscript has been checked by an English editing service, we will ask it again with this caution.

[reviewer comment] What does 'firn albedo setting' mean in the abstract? Perhaps explain briefly here.

[author reply] We changed here as "firn albedo, which is a fixed value in the model,".

[reviewer comment] It would be useful to provide some more detail about the nearby stations used in Table one – e.g. elevation, length of observations, not only distance but also direction from the ice core site.

[author reply] We provide the information of meteorological stations; location (Lon., Lat., elevation), starting year of observation, and direction from the corresponding ice core site as a supplementary table.

[reviewer comment] Can you split the data in figure 8 into four separate graphs on the one figure? It would be preferable to see the four sites more clearly. Another option would simply be to make the x axis far longer (although still covering the same temp interval) so that it is easier to differentiate the four sites. It is hard to see the orange and pink dots.

[author reply] In this figure, we want to show different relationships between ice layer thickness and reconstructed SMT so that separated graphs is not our preference. We made the x-axis 1.5 times longer than the present one and smaller the symbols to show red dots behind blue dots (shown as Fig. 1).

―――――――――――――――――――――――

[Figure]

**Fig. 1.** Modified figure of Fig. 8